# Structural and dynamic changes in P-Rex1 upon activation by PIP$_3$ and inhibition by IP$_4$

**Sandeep K Ravala[1], Sendi Rafael Adame-Garcia[2], Sheng Li[3], Chun-Liang Chen[1], Michael A Cianfrocco[4], J Silvio Gutkind[2], Jennifer N Cash[5]\*, John JG Tesmer[1]\***

[1]Departments of Biological Sciences and of Medicinal Chemistry and Molecular Pharmacology, Purdue University, West Lafayette, United States; [2]Department of Pharmacology and Moores Cancer Center, University of California, San Diego, San Diego, United States; [3]Department of Medicine, University of California, San Diego, La Jolla, United States; [4]Department of Biological Chemistry, University of Michigan, Ann Arbor, United States; [5]Department of Molecular and Cellular Biology, University of California, Davis, Davis, United States

**\*For correspondence:**
jcash@ucdavis.edu (JNC);
jtesmer@purdue.edu (JJGT)

**Competing interest:** The authors declare that no competing interests exist.

**Abstract** PIP$_3$-dependent Rac exchanger 1 (P-Rex1) is abundantly expressed in neutrophils and plays central roles in chemotaxis and cancer metastasis by serving as a guanine-nucleotide exchange factor (GEF) for Rac. The enzyme is synergistically activated by PIP$_3$ and heterotrimeric Gβγ subunits, but mechanistic details remain poorly understood. While investigating the regulation of P-Rex1 by PIP$_3$, we discovered that Ins(1,3,4,5)P$_4$ (IP$_4$) inhibits P-Rex1 activity and induces large decreases in backbone dynamics in diverse regions of the protein. Cryo-electron microscopy analysis of the P-Rex1·IP$_4$ complex revealed a conformation wherein the pleckstrin homology (PH) domain occludes the active site of the Dbl homology (DH) domain. This configuration is stabilized by interactions between the first DEP domain (DEP1) and the DH domain and between the PH domain and a 4-helix bundle (4HB) subdomain that extends from the C-terminal domain of P-Rex1. Disruption of the DH–DEP1 interface in a DH/PH-DEP1 fragment enhanced activity and led to a more extended conformation in solution, whereas mutations that constrain the occluded conformation led to decreased GEF activity. Variants of full-length P-Rex1 in which the DH–DEP1 and PH–4HB interfaces were disturbed exhibited enhanced activity during chemokine-induced cell migration, confirming that the observed structure represents the autoinhibited state in living cells. Interactions with PIP$_3$-containing liposomes led to disruption of these interfaces and increased dynamics protein-wide. Our results further suggest that inositol phosphates such as IP$_4$ help to inhibit basal P-Rex1 activity in neutrophils, similar to their inhibitory effects on phosphatidylinositol-3-kinase.

## eLife assessment

This **important** study contributes insights into the regulatory mechanisms of a protein governing cell migration at the membrane. The integration of approaches revealing protein structure and dynamics provides **convincing** data for a model of regulation and suggests a new allosteric role for a solubilized phospholipid headgroup. The work will be interesting to researchers focusing on signaling mechanisms, cell motility, and cancer metathesis.

## Introduction

Localized activation of signaling is required for proper cell migration. Phosphatidylinositol 3,4,5-trisphosphate ($PIP_3$)-dependent Rac exchanger 1 (P-Rex1) is a Rho guanine-nucleotide exchange factor (RhoGEF) abundantly expressed in neutrophils that mediates chemotaxis and the generation of reactive oxygen species via activation of Rac GTPases (*Dorseuil et al., 1992*). The protein is comprised of a catalytic Dbl homology (DH) domain followed by a pleckstrin homology (PH) domain, two DEP domains (DEP1 and DEP2), two PDZ domains (PDZ1 and PDZ2), and a C-terminal inositol polyphosphate-4-phosphatase-like (IP4P) domain (*Figure 1A*).

P-Rex1 exhibits low basal activity until it becomes activated via direct interaction with membrane-bound regulators $PIP_3$ and Gβγ which act synergistically (*Cash et al., 2019*; *Cash et al., 2016*; *Mayeenuddin et al., 2006*; *Welch et al., 2002*), indicating that they use distinct modes of regulation. Although relatively little is known about how P-Rex1 transitions to an activated state, recent structural studies have defined their docking sites. Gβγ engages a scaffold composed of an amalgamation of the DEP2-PDZ1-PDZ2-IP4P domains and likely helps recruit P-Rex1 to the cell membrane (*Cash et al., 2019*). In contrast, $PIP_3$ binds to the PH domain (*Hill et al., 2005*) in a basic pocket (*Cash et al., 2016*), but this is not necessary for its recruitment to the cell membrane, implying that $PIP_3$ instead induces a conformational change that activates the enzyme (*Cash et al., 2016*). Because domains C-terminal to the catalytic DH domain are well known to be involved in autoinhibition (*Chávez-Vargas et al., 2016*; *Hill et al., 2005*; *Ravala et al., 2020*; *Urano et al., 2008*), the allosteric change induced by $PIP_3$ must defeat interdomain contacts and render the catalytic DH domain accessible to its substrate.

Here, we used hydrogen-deuterium exchange mass spectrometry (HDX-MS), cryo-electron microscopy (cryo-EM), single-particle analysis (SPA), and small-angle X-ray scattering (SAXS) along with functional studies and live cell experiments to show that activation of P-Rex1 involves disruption of two different inhibitory interfaces between domains across the length of the protein. Surprisingly, we found that the $PIP_3$ headgroup analog $IP_4$ can reduce P-Rex1 activity by stabilizing the autoinhibited conformation of the enzyme at physiologically relevant concentrations, suggesting a previously unknown, additional mechanism of regulation. Our experiments further suggest that P-Rex1 binding to $PIP_3$-containing membranes induces conformational changes that unwind P-Rex1 into a fully active state.

## Results

### $IP_4$ induces protection from deuterium incorporation on regions of P-Rex1 distal from the $PIP_3$-binding site

Previous work suggested that $PIP_3$ binding to the PH domain activates P-Rex1 purely through an allosteric mechanism (*Cash et al., 2016*). Thus, we anticipated that binding of the soluble headgroup of $PIP_3$, $IP_4$, to full-length P-Rex1 could also lead to conformational changes characteristic of the activated state. To test this, we analyzed P-Rex1 in the presence and absence of $IP_4$ using HDX-MS. We observed strong protection from deuterium incorporation in the $PIP_3$-binding site on the PH domain in the presence of $IP_4$ (*Figure 1A and B*). However, we also observed strong protection in other regions of the protein: namely on the surface of the PH domain, particularly in the β5/β6 loop, and in several regions within an extension of the C-terminal IP4P domain that was not visualized in the P-Rex1–Gβγ complex (*Figure 1A and C*; *Cash et al., 2019*). We speculated that these diverse regions form more stable long-range interactions in the presence of $IP_4$.

### $IP_4$ allosterically inhibits P-Rex1

Based on our HDX-MS data, we hypothesized that $IP_4$ could inhibit activity of full-length P-Rex1. Using an in vitro GEF activity assay on soluble Cdc42 in the presence of liposomes, we observed that $IP_4$ inhibits $PIP_3$-mediated activation of P-Rex1 with an $IC_{50}$ value of 1.4 μM (*Figure 1D*). Competition was not observed with Ins(1,4,5)$P_3$, indicating that inhibition is dependent on the 3-phosphate, which is critical for $PIP_3$ binding to the P-Rex1 PH domain (*Cash et al., 2016*). However, $IP_4$ did not affect the activity of the P-Rex1 DH/PH or DH/PH-DEP1 fragments (*Figure 1—figure supplement 1A and B*). Collectively, these results indicated that $IP_4$ inhibits P-Rex1 allosterically and that this inhibition is dependent on long-range interactions between the regions shown to be protected by $IP_4$ in the Gβγ-binding scaffold (DEP2-PDZ1-PDZ2-IP4P) and in the DH/PH-DEP1 module. Negatively charged

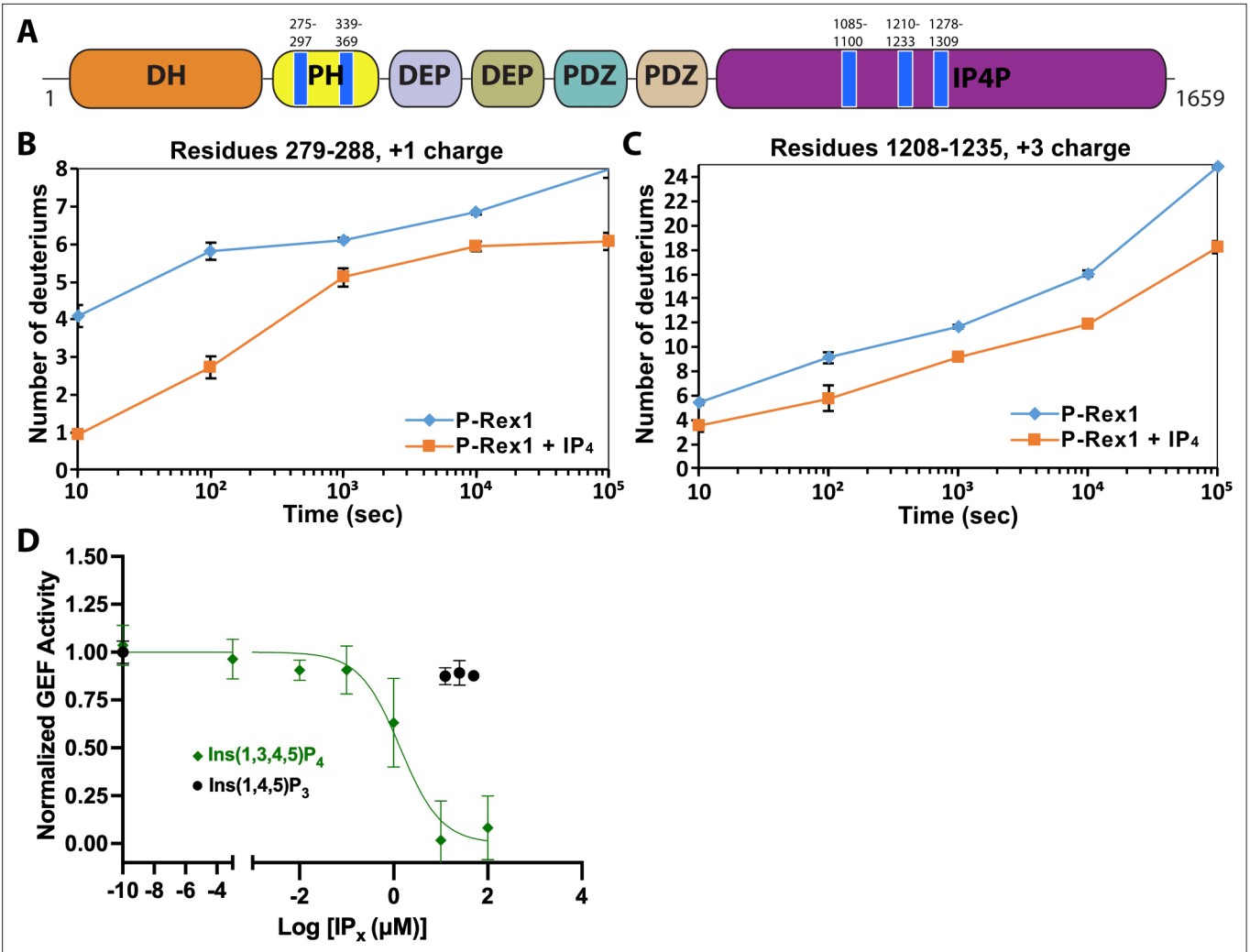

**Figure 1.** IP$_4$ binding causes dynamic changes in multiple domains of P-Rex1 and inhibits PIP$_3$-induced activation. (**A**) Difference hydrogen-deuterium exchange mass spectrometry (HDX-MS) data plotted onto the domain layout of P-Rex1. Blue regions indicate less deuterium uptake upon IP$_4$ binding. Graphs show the exchange over time for select regions in the P-Rex1 (**B**) PH domain and (**C**) an IP4P region that was disordered in the P-Rex1–Gβγ structure. The average of two experiments is plotted with the bars representing the range of each time point. (**D**) In vitro GEF activity of P-Rex1 evaluated on liposomes containing 2.5 µM PIP$_3$ in the presence of varying IP$_4$ concentrations (0–100 µM). Data were fit to exponentials to get rate constants by constraining the span to be shared. The resulting rates for each experiment were normalized by averaging two PIP$_3$ data points and two PC/PS data points to represent the top and bottom of the binding curve. The resulting normalized rates (min$^{-1}$) were fit with a one-phase binding curve wherein the top and bottom were constrained to 1 and 0, respectively, and the Hill coefficient fixed at –1. The resulting IC$_{50}$ was 1.4 µM with a confidence interval of 0.81–2.3. Data represent 4–5 independent experiments. Error bars represent the mean ± S.D.

The online version of this article includes the following figure supplement(s) for figure 1:

**Figure supplement 1.** P-Rex1 GEF activity assays show that truncated constructs are not inhibited by IP$_4$ and are not activated by PIP$_3$-containing liposomes.

liposomes (containing PC/PS), including those that also contain PIP$_3$, inhibit the GEF activity of the DH/PH-DEP1 and DH/PH fragments (*Figure 1—figure supplement 1C*). Because full-length P-Rex1 is not affected by PC/PS liposomes, this suggests that the observed inhibition represents a non-productive interaction of the DH/PH-DEP1 and DH/PH fragments with negatively charged surfaces in our assay. The lack of activation of DH/PH-DEP1 by PIP$_3$ prevents us from testing in this assay whether IP$_4$ can inhibit via direct competition with PIP$_3$ at the PH domain.

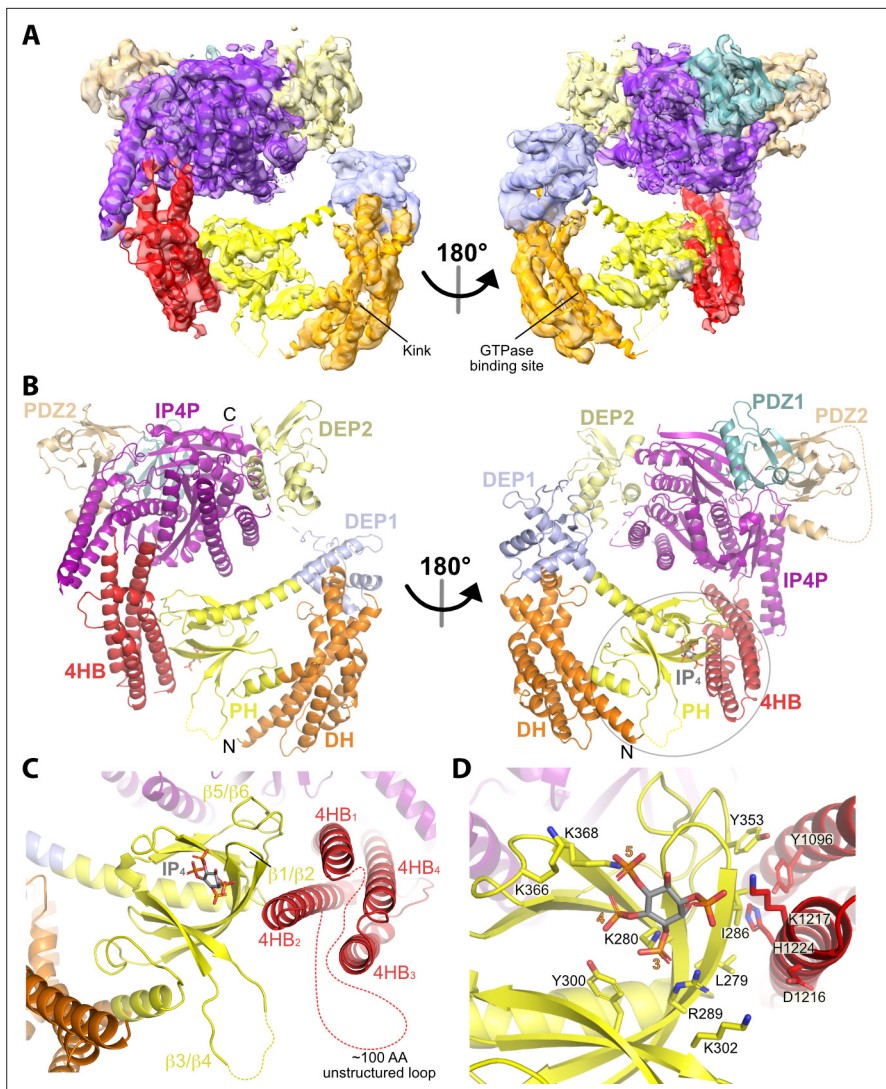

**Figure 2.** Structure of the P-Rex1·IP$_4$ complex in an autoinhibited conformation. (**A**) Cryo-EM reconstruction with atomic model superimposed. The kink between the DH and PH domains and the GTPase binding site is labeled. (**B**) Atomic model without the cryo-EM map. (**C**) The PH–4HB interface primarily involves the β1/β2 and β5/β6 loops of the PH domain, which were previously shown to be involved in protein-protein interactions in crystal structures (**Cash et al., 2016**), and the 4HB$_1$ and 4HB$_2$ helices of the 4HB domain. Flexible loops, including the basic β3/β4 loop of the PH domain involved in membrane binding (**Cash et al., 2016**), are shown as dashed lines. We speculate that this loop could interact with phosphorylated residues in the adjacent 4HB unstructured loop. (**D**) Side chains in the PH–4HB interface. The 3-, 4-, and 5-position phosphates of bound IP$_4$ are labeled. Note that PIP$_3$ could not bind to the PH domain in this state due to steric blockade by the 4HB domain. The area of focus in (**C**) and (**D**) is circled in gray in (**B**).

The online version of this article includes the following source data and figure supplement(s) for figure 2:

**Figure supplement 1.** Samples of P-Rex1 ± IP$_4$ imaged on a Glacios microscope highlight a conformation of P-Rex1 that is stabilized by IP$_4$.

**Figure supplement 1—source data 1.** Original SDS-PAGE gel.

**Figure supplement 2.** Cryo-EM 2D classification and 3D reconstruction.

**Figure supplement 3.** The overall structure of the P-Rex1·IP$_4$ complex is similar to that of unliganded P-Rex1.

# IP$_4$ stabilizes long-range interactions mediated by the P-Rex1 DEP1 and PH domains

To understand the molecular basis of IP$_4$-mediated stabilization and inhibition, we analyzed full-length P-Rex1 with and without IP$_4$ using cryo-EM SPA (*Figure 2—figure supplement 1A*). Initial datasets were collected using a Glacios transmission electron microscope and then processed through 2D classification (*Figure 2—figure supplement 1B–D*). In both datasets, most classes showed only the Gβγ-binding core of P-Rex1 (*Figure 2—figure supplement 1C and D*). A few classes contained particles with additional mass to the side of the core, close to the PDZ or DEP domains (*Figure 2—figure supplement 1C and D*, orange boxes). This mass could represent either the N-terminal domains or another P-Rex1 particle in proximity. Only in the sample containing IP$_4$ could we observe class averages with additional mass next to the core opposite the side that binds Gβγ (*Figure 2—figure supplement 1D*, green boxes). Based on its size, location, our HDX-MS data (*Figure 1*), and low-resolution maps of P-Rex1 generated in a previous study (*Cash et al., 2019*), this mass most likely corresponded to the N-terminal DH/PH-DEP1 domains interacting with an elongated subdomain extending from the IP4P domain.

We next collected much larger datasets on the P-Rex1·IP$_4$ complex using a Krios transmission electron microscope and determined the structure of this complex at an average resolution of 4.1 Å (*Figure 2A and B*, *Table 1*, and *Figure 2—figure supplement 2*). Similar to the P-Rex1–Gβγ complex (*Cash et al., 2019*), this sample exhibited a preferred orientation on grids, necessitating the addition of data collected on a tilted sample (*Figure 2—figure supplement 2A and C*). The resulting 3D reconstruction clearly shows the Gβγ-binding core, composed of DEP2, PDZ1, PDZ2, and the majority of the IP4P domain, from which there are two extensions of density that contact one another to form a loop-like structure (*Figure 2A*, *Figure 2—figure supplement 2D*). One extension corresponds to a large insertion in the IP4P domain that contains IP$_4$-stabilized regions (*Figure 1*) and that was disordered in the P-Rex1–Gβγ complex (*Cash et al., 2019*). The ordered elements of the insertion form a long 4-helix bundle (4HB) most similar in fold to focal adhesion targeting (FAT) domains (*Hayashi et al., 2002*), which are found in other peripheral membrane proteins involved in cell adhesion and migration. The other extension corresponds to the DH/PH-DEP1 domains. Based on its distinct shape, the PH domain (*Cash et al., 2016*) was fit into density along the side of the 4HB. Here, IP$_4$ is observed bound to the PH domain PIP$_3$-binding site (*Figure 2B–D*). The position of the DH domain was also obvious, but individual helices of the DH domain were lower in resolution and more dynamic relative to the rest of the structure (*Figure 2—figure supplement 2D*). The relative positions of the DH and PH domains mandate a severe bend in the helix connecting the DH and PH domains (*Figure 2A and B*), resulting in a jack-knifed configuration of the DH/PH module that blocks access to the GTPase binding site on the DH domain. The remaining mass, immediately adjacent to the end of the DH domain opposite its N-terminus, corresponds to DEP1. Using the required connectivity of its N- and C-termini to the PH and DEP2 domains, respectively, DEP1 was docked in a manner that complemented residues on the DH domain (*Figure 3A and B*). Weak density corresponding to a long five-turn extension of the αC helix of the PH domain connects to the N-terminus of DEP1, but its C-terminal connection to DEP2 is disordered, likely explaining lower local resolution in the DEP1 region (*Figure 2—figure supplement 2D*). Overall, the conformation of the P-Rex1 Gβγ-binding core is essentially the same as in the P-Rex1–Gβγ complex (RMSD deviation of 1.1 Å for 701 Cα atoms).

The contact between the PH domain and the 4HB is primarily mediated via the β1 and β2 strands and β5/β6 loop of the PH domain (*Figure 2C and D*). In all previous crystal structures including the P-Rex1 PH domain, this same surface formed extensive protein–protein lattice contacts (*Cash et al., 2016*; *Lucato et al., 2015*). The residues directly involved in the interface are among the most strongly protected in the presence of IP$_4$ as measured by HDX-MS (*Figure 1A–C*, *Figure 4A*). The PH domain β3/β4 loop, which we previously showed to be a nonspecific anionic membrane-binding loop, remains unstructured and is situated near a loop at the tip of the 4HB that is also unstructured (residues 1109–1209, *Figure 2C*) and contains known phosphorylation sites, some of which regulate activity (*Barber et al., 2012*). At the interface, surface hydrophobic residues Leu279 and Ile286 on the PH β1 and β2 strands and Tyr353 on the β5/β6 loop interact with a surface of 4HB including Tyr1096 and His1224 (*Figure 2D*). Charge complementarity is formed between Lys302 in the PH domain and Asp1216 in 4HB, and between the 1-phosphate of IP$_4$ and Lys1217. Otherwise, IP$_4$ does not make direct contact with the 4HB. However, because different PH domain ligands uniquely perturb the conformation of

**Table 1.** Cryo-EM data collection, refinement, and validation statistics.

**Structure: P-Rex1–IP$_4$**
**(EMDB: EMD-41621)**
**(PDB: 8TUA)**
**(EMPIAR: EMPIAR-11967)**

| | Untilted | Tilted |
|---|---|---|
| **Data collection** | | |
| Grids | Carbon Quantifoil | Carbon Quantifoil |
| Vitrification method | FEI Vitrobot | FEI Vitrobot |
| Microscope | Titan Krios | Titan Krios |
| Magnification | 81,000 | 81,000 |
| Voltage (kV) | 300 | 300 |
| Stage tilt (°) | 0 | 30 |
| Detector | K3 DED | K3 DED |
| Recording mode | Counting | Counting |
| Total electron exposure (e–/Å$^2$) | 57.8 | 57.8 |
| Number of frames | 40 | 40 |
| Defocus range (µm) | 0.2–2.0 | 0.2–2.0 |
| Pixel size (Å) | 1.054 | 1.054 |
| **Data processing** | | |
| Number of micrographs | 2127 | 3069 |
| Initial particle images (no.) | 806,067 | 1,620,545 |
| Final particle images (no.) | 89,450 | 119,739 |
| Initial particle images merged (no.) | 209,189 | |
| Final total particle images (no.) | 187,734 | |
| Symmetry | C1 | |
| Map resolution (Å) | 4.1 | |
| **Refinement** | | |
| Initial model used (PDB code) | 6PCV, 6VSK, 5D3X, 5FI1, 7RX9 | |
| Model resolution (Å) Fourier shell correlation threshold | 4.1 0.143 | |
| Map sharpening $B$ factor (Å$^2$) | –176 | |
| *Model composition* Non-hydrogen atoms Hydrogens Protein residues Ligands | 10,693 10,720 1,330 1 (4IP) | |
| *B factors (Å$^2$; min/max/mean)* Protein Ligand | 30.9/202/123 143/143/143 | |
| *R.m.s. deviations* Bond lengths (Å) *Bond angles* (°) | 0.003 0.505 | |
| *Validation* MolProbity score Clashscore Rotamer outliers (%) CaBLAM outliers (%) | 2.03 13.88 0.94 2.2 | |

*Table 1 continued on next page*

*Table 1 continued*

**Structure: P-Rex1–IP$_4$**
**(EMDB: EMD-41621)**
**(PDB: 8TUA)**
**(EMPIAR: EMPIAR-11967)**

| *Ramachandran plot* (%) | |
|---|---|
| Favored | 94.5 |
| Allowed | 5.5 |
| Outliers | 0 |

| *Model vs. data* | |
|---|---|
| CC mask | 0.67 |
| CC box | 0.73 |
| CC peaks | 0.58 |
| CC volume | 0.66 |
| Mean CC for ligand | 0.67 |

these regions in the PH domain (*Cash et al., 2016*), IP$_4$ could indirectly stabilize this interface by trapping the β1 and β2 strands and β5/β6 loop in a conformation with higher affinity for 4HB.

The jack-knife in the helix between the DH and PH domains is required to allow the PH domain to interact with the 4HB and is stabilized by the DEP1 domain docking to the DH domain (*Figure 2B*). DH domain residues Leu173, Leu177, and Leu178 form a hydrophobic interface with DEP1 residues Ile409, Ile457, Leu466, and Ala469 (*Figure 3A*). Leu173 and Leu177 were previously noted to be conspicuously exposed in structures of the DH/PH tandem (*Cash et al., 2016*). Thus, the DEP1 domain stabilizes an inactive DH/PH tandem that is further stabilized by interaction with the 4HB of the IP4P domain.

While conducting these studies, the structure of human P-Rex1 in the absence of IP$_4$ was reported (PDB entry 7SYF) (*Chang et al., 2022*), allowing a comparison between the IP$_4$ and IP$_4$-free states of autoinhibited P-Rex1 (*Figure 2—figure supplement 3*). Overall, the domain organization is very similar, but there is an ~3° rotation of the Gβγ-binding core in the IP$_4$ complex relative to the PH–4HB interface such that DEP1 and DEP2 move closer together. It is possible that the binding of IP$_4$ at this interface drives this conformational change. A caveat is that the protein used for the 7SYF structure contained a T4 lysozyme domain inserted into the β3/β4 loop of the PH domain. Although this domain was not visible in the reconstruction, its proximity to the IP$_4$ binding site may influence the global conformation of P-Rex1. The similarities between P-Rex1 ± IP$_4$ also suggest that the IP$_4$-binding site in the PH domain is freely accessible in the autoinhibited state. PIP$_3$ would, however, not have access because the 4HB domain would block binding to a membrane surface.

## The DH–DEP1 interface contributes to autoinhibition in vitro

To test the contribution of the observed DH–DEP1 interface to autoinhibition, the interface was disrupted by site-directed mutagenesis in the context of the DH/PH-DEP1 fragment. This fragment is five- to tenfold less active (depending on assay conditions) than the DH/PH tandem alone (*Figure 1—figure supplement 1C*; *Ravala et al., 2020*), confirming a specific role for DEP1 in autoinhibition (*Figure 3C*). Single-point mutations in the interface profoundly affected GEF activity on soluble Cdc42. The L173A variant had an ~2.5-fold higher activity, whereas L177E and L178E exhibited four- to fivefold higher activity, similar to the activity of DH/PH without the DEP1 domain. In the DEP1 domain, I409A and L466A mutations resulted in approximately two- and fourfold higher activity, respectively. The α1/α2 loop (residues 77–90) of the DH domain, although ordered in previous structures of DH/PH bound to GTPases (*Cash et al., 2016*; *Lucato et al., 2015*), is disordered in our structure (*Figure 3B*). However, modeling the DH domain from previous DH/PH crystal structures suggests that Lys89 and Arg78 would be close enough to form a bipartite ionic interaction with Glu456 in the DEP1 domain. Consistent with this hypothesis, the E456K variant was ~2.5-fold more active. Surprisingly, one of the intended disruptive mutations, A170K, instead inhibited GEF activity by ~50% (*Figure 3C*). Based on the available structures, Lys170 in the DH domain could form complementary electrostatic interactions with Glu411 along with additional non-polar interactions with the backbone and side chains of residues 409, 411, and 412 in the DEP1 domain, thereby stabilizing the DH–DEP1 interface (*Figure 3B*).

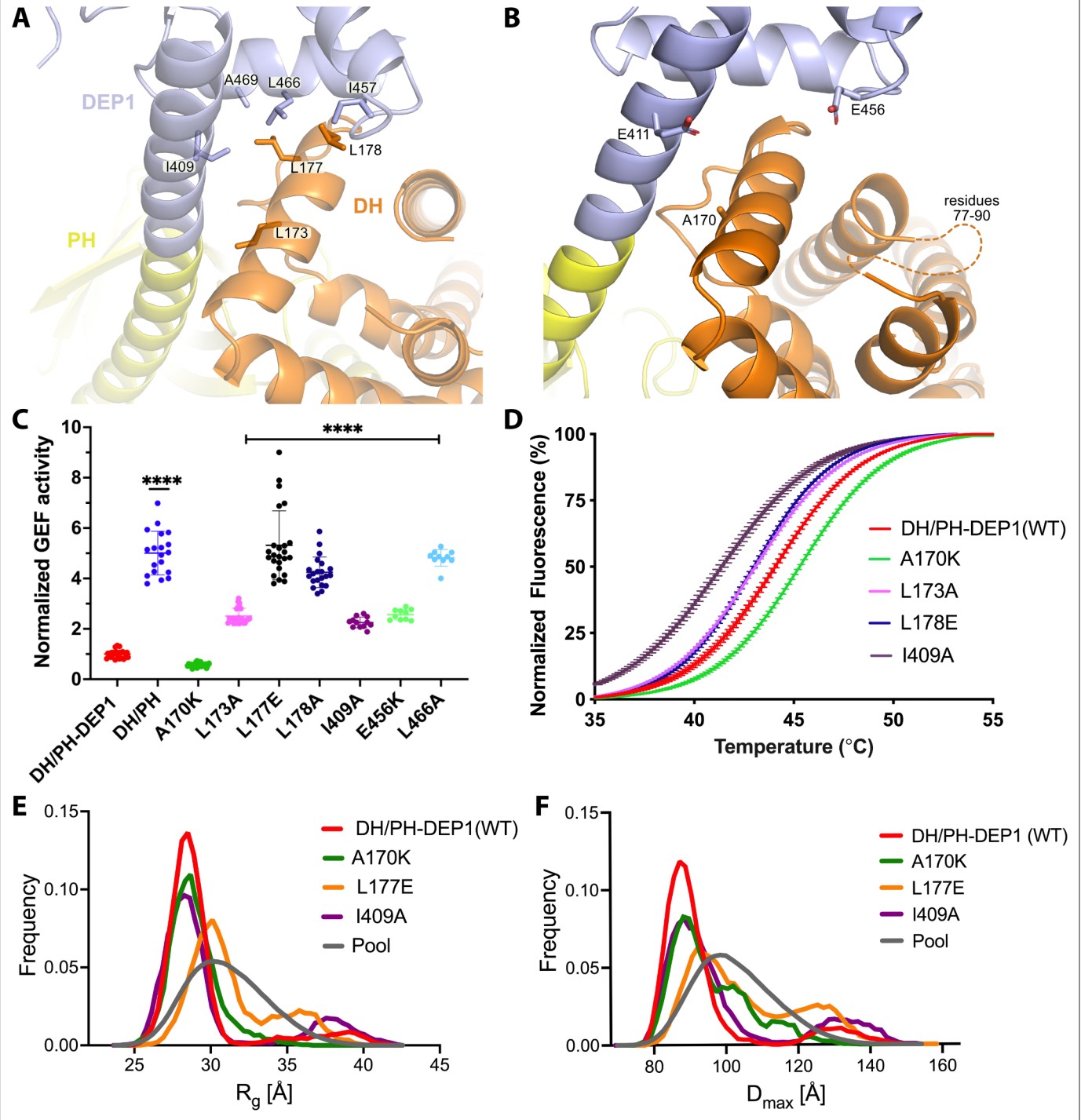

**Figure 3.** Mutations at the DH–DEP1 interface alter stability, conformation, and activity of DH/PH-DEP1. (**A**) Side chains that contribute to the hydrophobic interface formed between the DH and DEP1 domains. (**B**) Electrostatic interactions contributing the DH–DEP1 interface. The dotted line indicates a disordered region on the DH domain containing positively charged residues that may interact with Glu456. The A170K mutant is expected to form a salt bridge with Glu411 and strengthen the interface. (**C**) Fluorescence-based in vitro GEF activity assay on soluble Cdc42 with variants of the purified DH/PH-DEP1 fragment. GEF activity in this experiment was fit to a one-phase exponential decay normalized to that of DH/PH-DEP1 (WT). ****p<0.0001. (**D**) Representative ThermoFluor analyses showing that mutations that disrupt the DH–DEP1 interface also destabilize the protein, as evidenced by decreased $T_m$ values for each variant (see **Table 2**). Data are normalized from 0 to 100% representing lowest and highest fluorescence values. Note that A170K, which inhibits activity in panel (**C**), increases stability. (**E, F**) Ensemble optimization method (EOM) analysis of small-angle X-ray scattering (SAXS) data collected from mutations disrupting the DH–DEP1 interface indicate that these variants exhibit more extended conformations

*Figure 3 continued on next page*

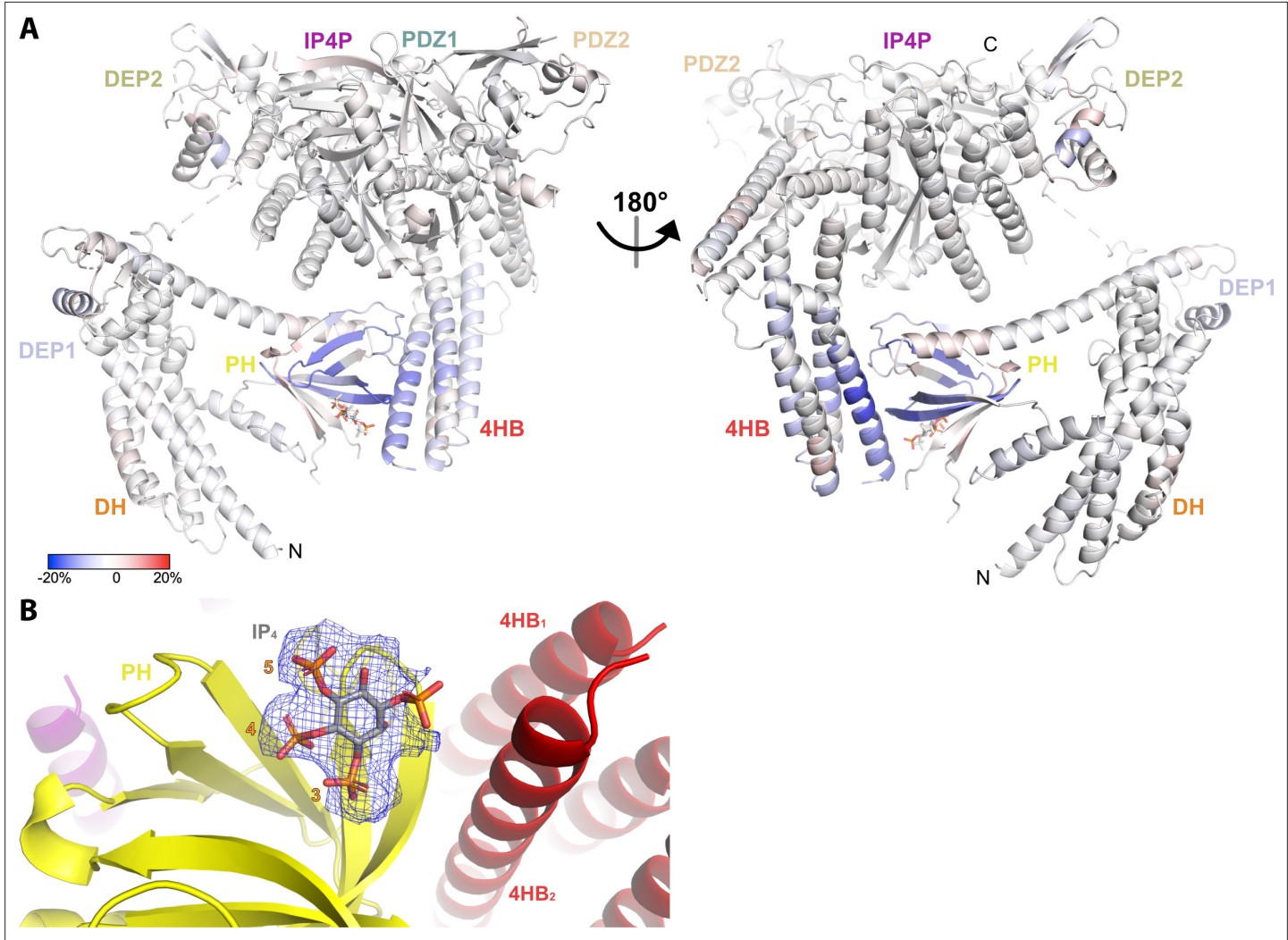

**Figure 4.** Hydrogen-deuterium exchange mass spectrometry (HDX-MS) and cryo-EM data support the conclusion that IP$_4$ stabilizes a closed conformation of P-Rex1. (**A**) Difference HDX-MS data plotted onto the structure of the P-Rex1 bound to IP$_4$. Blue regions indicate more protection upon IP$_4$ binding, whereas red regions indicate less. See also *Figure 4—source data 1*. (**B**) Map representing IP$_4$ bound in the PIP$_3$-binding site of the PH domain. The 3-, 4-, and 5-phosphates of IP$_4$ are reasonably well-ordered.

The online version of this article includes the following source data for figure 4:

**Source data 1.** HDX-MS data on P-Rex1–IP$_4$ with time points.

## The DH–DEP1 interface stabilizes DH/PH-DEP1 and decreases flexibility

Because disruption of the DH–DEP1 interface led to increased activity, we predicted that these variants would have a more extended DH/PH module that is likely more dynamic. To test this, wild-type (WT) DH/PH-DEP1 and its variants were assessed using a ThermoFluor assay to determine their melting temperatures (T$_m$). Indeed, variants with increased activity also had lower T$_m$ values (*Figure 3D*, *Table 2*). Conversely, A170K, which was less active than WT, showed a higher T$_m$. We also analyzed these variants using size-exclusion chromatography coupled to small-angle X-ray scattering (SEC-SAXS) (*Figure 3—figure supplement 2A and B*). In previous SAXS analyses, we observed that DH/PH-DEP1 exhibited a more compact state with a smaller conformational landscape in solution relative to DH/PH (*Ravala et al., 2020*). We hypothesized that mutations which disrupt the DH–DEP1 interface would likewise lead to more elongated ensembles. Compared to WT DH/PH-DEP1, which had a radius of gyration (R$_g$) of 30 ± 0.3 Å, the L177E variant (the most active DH/PH-DEP1 variant tested) had an R$_g$ of 31 ± 0.2 Å, suggestive of expansion (*Figure 3E*, *Table 3*). Variants A170K and I409A

**Table 2.** ThermoFluor measurements of DH/PH-DEP1 variants.

| DH/PH-DEP1 | $T_m$ (°C) |
|---|---|
| WT | 44.0 ± 0.2 |
| A170K | 45.4 ± 0.3 (p<0.0001) |
| L173A | 43.0 ± 0.06 (p<0.0001) |
| L177A | 43.4 ± 0.4 (p=0.0076) |
| L177E | 43.4 ± 0.3 (p=0.0041) |
| L178A | 43.1 ± 0.1 (p<0.0001) |
| L178E | 42.9 ± 0.1 (p<0.0001) |
| I409A | 41.7 ± 0.8 (p<0.0001) |
| E411K | 43.9 ± 0.1 ns |
| K415A | 43.2 ± 0.5 (p=0.0047) |
| L451A | 41.3 ± 0.5 (p<0.0001) |
| E456K | ND |
| L466A | ND |

From two independent experiments performed in triplicate.

p-values are from one-way ANOVA comparisons with WT.

ND = not determined because inflection point not observed; ns = not significant.

**Table 3.** SAXS parameters for DH/PH-DEP1 variants.

| | WT | A170K | L177E | I409A |
|---|---|---|---|---|
| **Guinier analysis** | | | | |
| I(0)* | 0.0081 ± 0.00004 | 0.0021 ± 0.003 | 0.0092 ± 0.00004 | 0.0034 ± 0.00002 |
| $R_g$ (Å) | 30 ± 0.3 | 29 ± 0.08 | 31 ± 0.02 | 30 ± 0.4 |
| $Q_{min}$ (Å$^{-1}$) | 0.0047 | 0.005 | 0.0047 | 0.0047 |
| $Q_{max}$ (Å$^{-1}$) | 0.353 | 0.353 | 0.353 | 0.353 |
| **P(r) analysis** | | | | |
| $D_{max}$ (Å) | 97 | 90 | 110 | 104 |
| Volume (Å$^3$) | 73,900 | 75,000 | 72,500 | 74,900 |
| $MM_{exp}$($MM_{cal}$) (kDa) | 54 (54) | 56 (54) | 54 (54) | 55 (55) |
| **EOM analysis** | | | | |
| Crystal structures | 5FI1;6VSK | 5FI1;6VSK | 5FI1;6VSK | 5FI1;6VSK |
| q-range (Å$^{-1}$) | .00475–0.353 | 0.00446–0.353 | .00475–0.353 | 00475–0.353 |
| $R_{flex}$ | 70.1% (82.6%) | 71.9% (85.3%) | 82.9% (84.9%) | 79.1% (84.9%) |
| $R_\sigma$ | 1.18 | 0.60 | 1.08 | 1.56 |
| Skewness | 2.39/0.41 | 1.11/0.40 | 0.85/0.42 | 1.50/0.40 |
| Kurtosis | 4.86/–0.08 | 2.09/–0.14 | –0.27/–0.12 | 0.77/–0.14 |

SAXS = small-angle X-ray scattering; EOM = ensemble optimization method.

*SAXS parameters I(0), $R_g$, $D_{max}$, $q_{min}$, $q_{max}$, MMexp, MMcal, $R_{flex}$, and $R_\sigma$ are the experimentally determined intensity at zero scattering angle, radius of gyration, maximum particle dimension, minimum scattering angle, maximum scattering angle, molecular mass calculated from scattering data, molecular mass calculated based on amino acid sequence, flexibility metric of ensemble in comparison (pool value in parentheses), and ratio of standard deviation for the distribution of selected ensemble to that of pool, respectively. The values for EOM analysis are from the last run of the genetic algorithm.

*Figure 3 continued*

(see *Table 3*). EOM analyses provide the $R_g$ and $D_{max}$ distributions derived from selected ensembles. The gray curves correspond to the $R_g$ and $D_{max}$ distributions for the pool of structures used for each analysis.

The online version of this article includes the following figure supplement(s) for figure 3:

**Figure supplement 1.** Introduction of a disulfide bridge in the DH/PH hinge reduces DH/PH-DEP1 activity.

**Figure supplement 2.** Small-angle X-ray scattering (SAXS) analyses of DH/PH-DEP1 variants.

had $R_g$ values similar to that of WT. Kratky plots indicated that all the samples had heterogeneous conformations (*Figure 3—figure supplement 2C*). The shapes of the P(r) functions were similar for all variants except a longer tail for L177E and I409A, consistent with a higher proportion of extended conformations in solution (*Figure 3—figure supplement 2D*).

Because the samples exhibited a high degree of heterogeneity in solution, the conformational distribution of these variants was assessed using the ensemble optimization method (EOM) (*Figure 3E and F*, *Figure 3—figure supplement 2A*; *Tria et al., 2015*). The resulting ensemble for WT shows predominant conformations with $R_g$ values ~28 Å and a small fraction of extended conformations with $R_g$ ~39 Å (*Figure 3E*). The selected ensemble for A170K had $R_g$ values similar to WT (~29 Å); however, the A170K peak is broader in comparison, suggesting conformational heterogeneity and structural changes. The L177E variant exhibited a larger shift to higher $R_g$, with an average $R_g$ ~30 Å, and a second significant population with $R_g$ >32 Å (*Figure 3E*). Similar to the distribution of $R_g$ values, the $D_{max}$ function distribution shows that the L177E variant had the most extended conformation of the variants tested (*Figure 3F*).

## Flexibility of the hinge in the α6-αN helix of the DH/PH module is important for autoinhibition

One of our initial goals in this project was to determine a high-resolution structure of the autoinhibited DH/PH-DEP1 core by X-ray crystallography. To this end, we started with the DH/PH-DEP1 A170K variant, which was more inhibited than wild-type but still dynamic, and then introduced S235C/M244C and K207C/E251C double mutants to completely constrain the hinge in the α6-αN helix via disulfide bond formation in a redox-sensitive manner. Single cysteine variants K207C and M244C were generated as controls. The S235C/M244C variant performed as expected, decreasing the activity of the A170K variant to nearly background in the oxidized but not the reduced state (*Figure 3—figure supplement 1*). However, the M244C single mutant exhibited similar effects, suggesting that it forms disulfide bonds with cysteine(s) other than S235C. Indeed, the side chains of Cys200 and Cys234 are very close to that of M244C. The K207C/E251C mutant was similar to S235C/M244C under oxidized conditions, but ~15-fold more active (similar to WT DH/PH levels, see *Figure 3C*) under reducing conditions. The K270C variant, on the other hand, exhibited higher activity than A170K on its own under oxidizing conditions, but similar activity to all the variants except K207C/E251C when reduced. These results suggest that K207C/E251C in a reduced state and K270C in an oxidized state favor a configuration where the DEP1 domain is less able to engage the DH domain and maintain the kinked state. The mechanism for this is not known. Regardless, these data show that perturbation of contacts between the kinked segments of the α6-αN helix can have profound consequences on the activity of the DH/PH-DEP1 core.

## Interactions at the P-Rex1 DH–DEP1 and PH–4HB interfaces contribute to autoinhibition in cells

To evaluate the roles of the P-Rex1 DH–DEP1 and PH–4HB interfaces in living cells, we utilized SRE luciferase-gene reporter assays as a read out of full-length P-Rex1 activity in HEK293T cells. Mutations at the DH–DEP1 interface had a strong effect on activity, with L177E and L466E (*Figure 3A*) exhibiting approximately tenfold and fourfold higher activity relative to WT, respectively (*Figure 5A*). Perturbation of the PH–4HB interface (*Figure 2D*) also increased activity in that Y1096A was approximately fivefold more active than WT (*Figure 5A*). Mutation of other residues in the PH–4HB interface (*Figure 2D*) also increased activity, although to a lesser extent.

The most affected variants (L177E, L466E, and Y1096A) were then tested for their effects on cell migration in response to chemokine gradients (*Figure 5B and C*, *Figure 5—figure supplement*

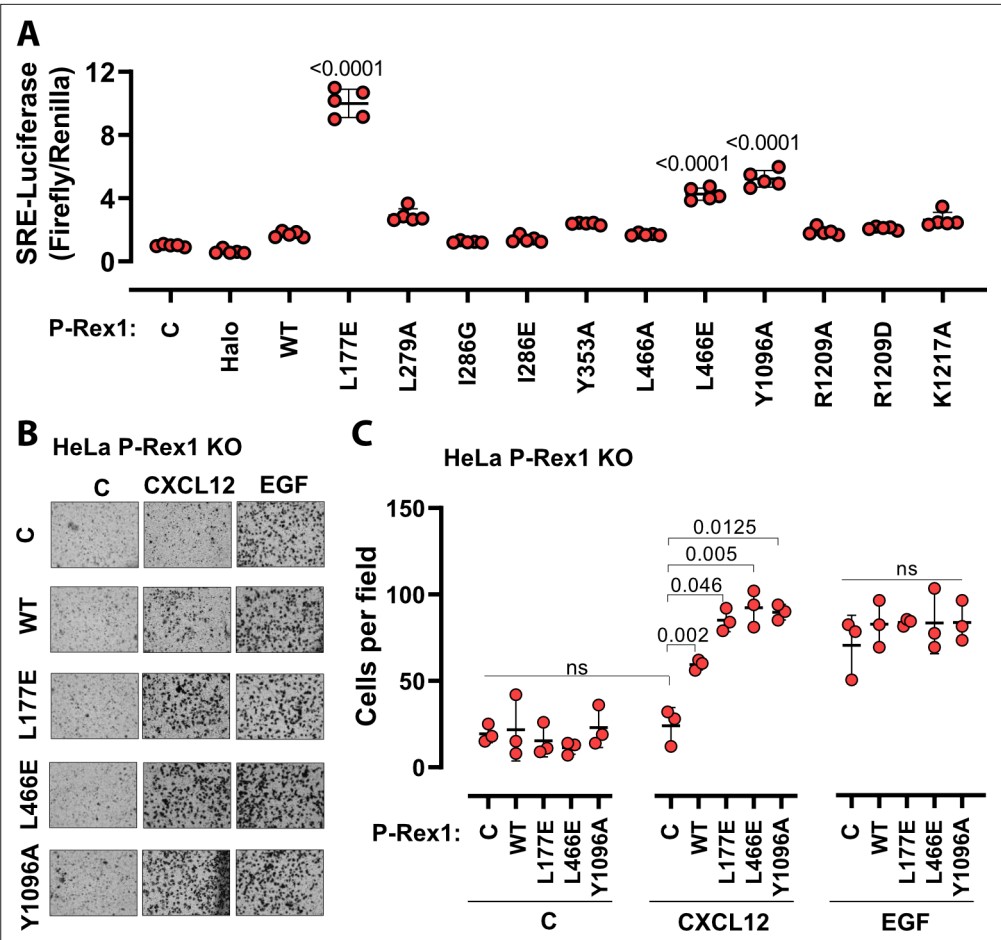

**Figure 5.** Disruption of the DH–DEP1 and PH–4HB interfaces leads to increased P-Rex1 activity in cells. (**A**) SRE luciferase-gene reporter assays. Mutations were cloned into full-length P-Rex1 in the pCEFL-HA-HaloTag vector, and these constructs, along with luciferase reporter genes, were co-transfected into HEK293T cells. Results depicted here are representative of three independent experiments, and error bars represent S.D. Non-transfected control (C) and empty vector transfected control (Halo) are shown. (**B, C**) Mutations which led to enhanced P-Rex1 activity in luciferase reporter assays were evaluated for their effect on chemotaxis of HeLa cells with endogenous P-Rex1 knocked out (HeLa P-Rex1 KO; see *Figure 5—figure supplement 1*). P-Rex1 constructs were transfected into HeLa P-Rex1 KO cells, and cell migration was evaluated in a trans-well migration assay upon stimulation with CXCL12 (50 ng/ml) or EGF (50 ng/ml). Data is presented as mean ± S.D. Significance (brackets) was determined using multiple comparison ANOVA followed by Šidák statistic test.

The online version of this article includes the following source data and figure supplement(s) for figure 5:

**Figure supplement 1.** P-Rex1 is required for CXCL12-stimulated cell migration downstream of the CXCR4 chemokine receptor.

**Figure supplement 1—source data 1.** Raw images of western blots.

---

*1A and B*). For this, the endogenous P-Rex1 gene in HeLa cells was first knocked out by CRIS-PR-Cas9 (*Figure 5—figure supplement 1C*) and the resulting cells were transfected with various P-Rex1 constructs (*Figure 5—figure supplement 1D*). Cell migration was then evaluated in the presence or absence of CXCL12 (upstream of P-Rex1 and Gβγ signaling) and epidermal growth factor (EGF). CXCL12-induced chemotaxis was dependent on the expression of P-Rex1 (*Figure 5B and C*, *Figure 5—figure supplement 1*), and all three variants caused a significantly larger number of cells to migrate. However, EGF-induced chemotaxis in HeLa cells, which is not dependent on P-Rex1, was unaffected by P-Rex1 expression. These data support that the DH–DEP1 and PH–4HB interfaces of P-Rex1 mediate autoinhibition that specifically modulates chemokine-induced cell migration.

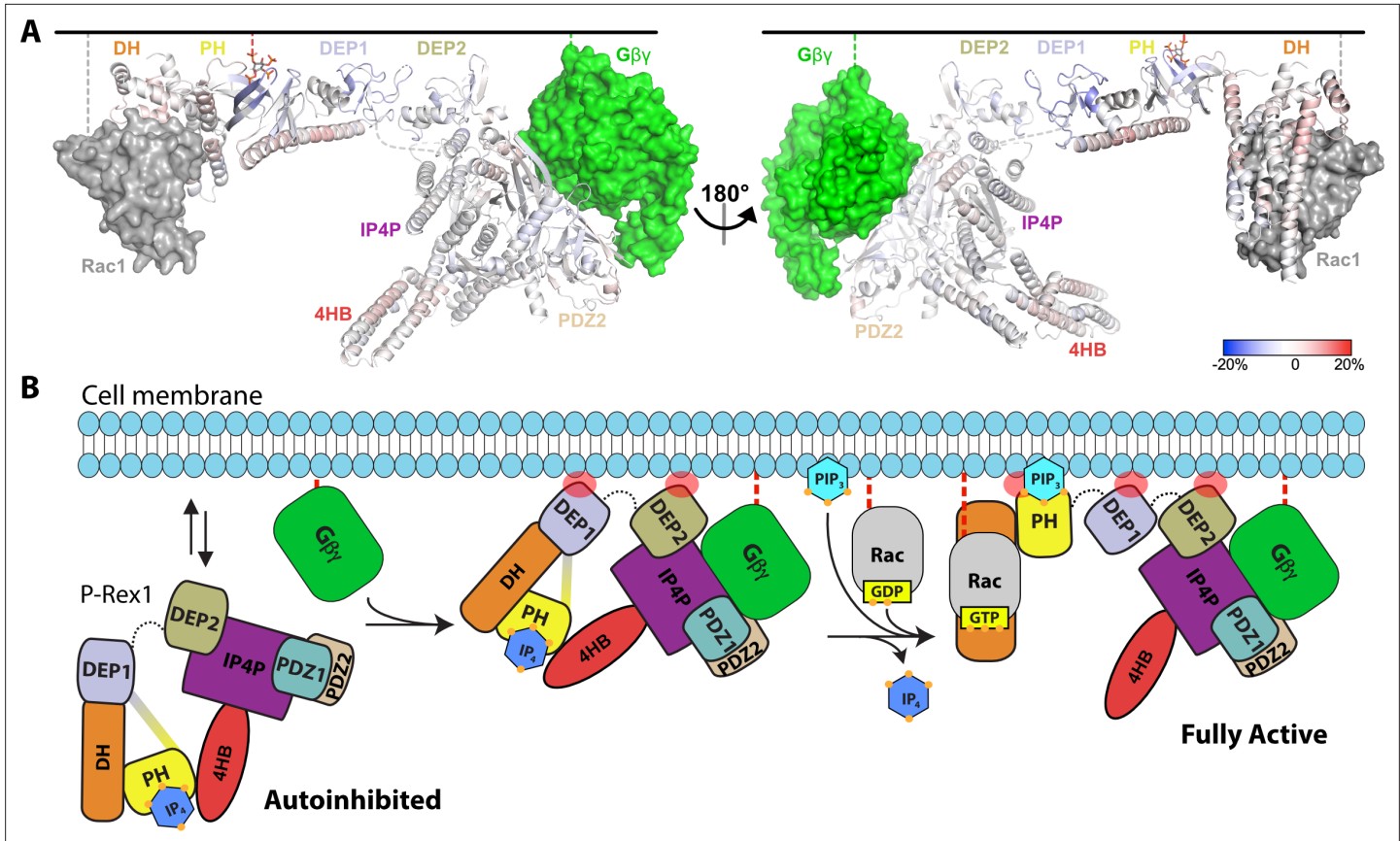

**Figure 6.** Hydrogen-deuterium exchange mass spectrometry (HDX-MS) supports that P-Rex1 undergoes long-range conformational changes when binding PIP$_3$-containing liposomes. (**A**) HDX-MS of P-Rex1 in the presence of PIP$_3$-containing liposomes. A model of P-Rex1 in an open conformation bound to a membrane containing PIP$_3$ was created and is shown colored according to difference HDX-MS data plotted onto the coordinates. HDX-MS data were collected in the presence of liposomes containing PIP$_3$ and compared to data collected on P-Rex1 alone. Blue and red regions indicate less and more protection, respectively, upon PIP$_3$-containing liposome binding. These changes occur specifically in the presence of PIP$_3$. See also *Figure 6—source data 1*. The black line at the top represents a membrane surface and the dashed lines represent covalent lipid modifications. Using available structural information, Gβγ and Rac1 were docked into this model (although neither were present in this HDX-MS experiment). (**B**) Cartoon schematic of our model of the steps involved in the activation of P-Rex1.

The online version of this article includes the following source data and figure supplement(s) for figure 6:

**Source data 1.** HDX-MS data on P-Rex1 plus liposomes with time points.

**Figure supplement 1.** Regions across the length of P-Rex1 become more exposed upon binding liposomes specifically containing PIP$_3$.

## P-Rex1 binding to PIP$_3$-containing model membranes induces a more open, dynamic conformation

Given the known binding site for Gβγ (*Cash et al., 2019*), the position of membrane-binding loops such as the β1/β2 loop of the DEP1 domain (*Ravala et al., 2020*), and the position of the 4HB domain, it does not seem that the PH domain would be able to interact with PIP$_3$ in a cell membrane while P-Rex1 is in its autoinhibited conformation. To better understand the molecular consequences of PIP$_3$ binding, HDX-MS measurements were taken on P-Rex1 in the presence of liposomes ± PIP$_3$. Without PIP$_3$, the most notable changes in P-Rex1 in response to liposomes were increases in protection of the lipid binding elements of both DEP domains (primarily their β1/β2 loops; *Figure 6—figure supplement 1A*), which suggests that they directly interact with lipid bilayers. Indeed, the isolated DEP1 can independently bind negatively charged liposomes (*Ravala et al., 2020*). Within the PH domain, there was deprotection of the αC helix, but there was no deprotection at the PH–4HB interface, suggesting that it remained intact. In contrast, membranes with PIP$_3$ caused deprotection of the entire DH domain, the α6-αN linker and the helices at the DH–DEP1 interface (*Figure 6A*, *Figure 6—figure supplement 1B*). Although the core of the PH domain showed, as expected, an increase in protection

in the presence of PIP$_3$, the structural elements contacting 4HB became deprotected, as did the PH-binding regions of 4HB. Collectively, these data are consistent with loosening of interdomain contacts and unraveling of at least some fraction of P-Rex1 onto the surface of the liposome.

## Discussion

Here, we showed that IP$_4$ binding to the PIP$_3$-binding site in full-length P-Rex1 stabilizes a closed, auto-inhibited conformation of P-Rex1 by enhancing long-range contacts across the length of the protein. DH–DEP1 and PH–4HB interactions are relieved upon P-Rex1 binding to PIP$_3$-containing liposomes, leading to its activation. Canonically, PIP$_3$ signaling is negatively regulated by PIP$_3$ phosphatases such as PTEN. Additionally, inositol phosphates, including IP$_7$ and IP$_4$, can compete with PIP$_3$ binding to PH domains (*Jia et al., 2007*), representing another form of negative regulation. Ins(1,3,4,5)P$_4$ is a major isoform of IP$_4$ in neutrophils (*Stuart et al., 1994*) where its concentration is estimated to be 4 µM (*French et al., 1991*) and where P-Rex1 is highly expressed (*Welch et al., 2002*). Because we can measure robust inhibition of P-Rex1 by IP$_4$ at >1 µM (*Figure 1D*), IP$_4$ may exert biologically relevant control of Rac activation, at least in neutrophils. Because P-Rex1 can bind membranes in the absence of PIP$_3$ or Gβγ, IP$_4$ may also serve to suppress P-Rex1 activity until a threshold concentration of PIP$_3$ is generated, allowing rapid activation of already membrane-associated P-Rex1. It is worth noting that such regulation by specific inositol phosphates may depend on the intracellular distribution of the enzymes responsible for their synthesis (*Gokhale et al., 2011*).

Another key result of our study was to provide a molecular explanation for how the PH and DEP1 domains contribute to P-Rex1 regulation. Nearly two decades ago, it was first reported that domains located C-terminal to the DH domain contribute to P-Rex1 autoinhibition (*Hill et al., 2005*). Deletion of the PH domain in P-Rex1 resulted in a large increase in activity in the context of the full-length enzyme. However, the DH/PH fragment has higher activity than DH/PH-DEP1 and larger P-Rex1 fragments, indicating that the PH domain itself is not intrinsically inhibitory, as it is in some other RhoGEF DH/PH tandems (*Bandekar et al., 2019*; *Chen et al., 2020*; *Ravala and Tesmer, 2024*). This apparent contradiction can now be explained by the fact that in DH/PH-DEP1 and larger fragments, the DH/PH module can jack-knife and position the PH domain in a manner that blocks GTPase binding. This model of regulation was corroborated by recent structural studies of P-Rex1 without IP$_4$ (*Chang et al., 2022*). The isolated DEP1 domain was also shown to play an autoinhibitory role based on the relatively low activity of DH/PH-DEP1 relative to DH/PH (*Ravala et al., 2020*). This can now be explained by its interaction with the DH domain, which, consequently, positions the PH domain to block access of GTPases. Because the PH and DEP1 domains can mediate inhibition in the contexts of both DH/PH-DEP1 and the full-length enzyme, and because there is little GEF activity when the DH–DEP1 interface is intact, the DH/PH-DEP1 module can be thought of as the core signaling circuit in P-Rex1. In support of this, our mutations in the DH–DEP1 interface were, in general, more activating than those in the PH–4HB interface in cells, although we note that our mutagenesis was not exhaustive (*Figure 5*).

Using our functional data along with the known structures of P-Rex1, we assembled a model for the activation of P-Rex1 by PIP$_3$ and Gβγ (*Figure 6B*). P-Rex1 in its basal state may be associated with IP$_4$ and exist in equilibrium between the membrane and the cytosol. Indeed, in our HDX-MS studies, the DEP1 and DEP2 domains show protection in the presence of liposomes without PIP$_3$ (*Figure 6—figure supplement 1*). In its autoinhibited configuration, the known membrane anchoring elements (the GTPase binding site of the DH domain, the PIP$_3$-binding site of the PH domain, the β3/β4 loop of the PH domain, the β1/β2 loops of DEP1 and DEP2, and the Gβγ-binding site in the C-terminal domains) cannot engage a common membrane plane (*Figure 6B*). The clear outlier is the PH domain that would not be able to engage the membrane along with the other domains without unwrapping of the autoinhibited conformation. PIP$_3$ is not a major driver of membrane anchoring on its own (*Barber et al., 2007*; *Cash et al., 2016*), and nor is Gβγ, but they do so synergistically in combination (*Barber et al., 2007*). In prior HDX-MS studies, Gβγ-binding did not have a large effect on the regions now known to be involved in autoinhibition and only caused protection in regions of direct contact (*Cash et al., 2019*). This supports the idea that its role in activation may be primarily related to translocation. This is consistent with the observation that the ΔPH variant of P-Rex1 (which cannot form the DEP1–DH or PH–4HB interfaces or bind PIP$_3$) is activated to the same extent by Gβγ as WT P-Rex1

(**Hill et al., 2005**). Furthermore, in autoinhibited P-Rex1, the Gβγ-binding site is readily accessible whereas that of $PIP_3$ is sequestered.

We propose that, after stimulation of GPCRs in neutrophils, Gβγ likely binds first and, with the assistance of membrane binding elements in the DEP1 and 2 domains and possibly the β3/β4 loop of the PH domain, promotes loosening of the autoinhibited state (**Figure 6B**). Generation of $PIP_3$ by PI3K then releases the DH/PH module from the 4HB and DEP1 domain, displacing any bound $IP_4$. Because neither $PIP_3$ nor $IP_4$ can activate DH/PH-DEP1 GEF activity on a soluble GTPase (**Figure 1—figure supplement 1**), we speculate that this unwrapping at the membrane with multiple points of engagement across the protein is necessary for full activation of P-Rex1 (**Figure 6B**). Indeed, our HDX-MS and SAXS studies here support that fully activated P-Rex1 at the membrane will be more extended and dynamic, rendering the DH domain more accessible to Rac1. However, what remains unknown is the mechanism by which the PH domain is able to access $PIP_3$ at the membrane, even in a 'loosened' autoinhibited state.

Additional layers of P-Rex1 regulation exist that remain underexplored. For example, phosphorylation of the lipid binding loop of DEP1 by PKA is known to inhibit P-Rex1 (**Chávez-Vargas et al., 2016**; **Ravala et al., 2020**). Also, a potential interaction may occur between the basic β3/β4 loop of the PH domain and the loop at the end of the 4HB (**Figure 2C**). Both loops, consistent with their extended and dynamic nature, have predicted and confirmed phosphorylation sites (**Barber et al., 2012**) and thus could potentially modulate P-Rex1 activity if they interact. Phosphorylation of the basic β3/β4 loop might be expected to inhibit activity based on the fact that it binds and localizes the protein to the negatively charged plasma membrane, consistent with dephosphorylation of the loop leading to activation (**Montero et al., 2016**; **Montero et al., 2013**).

## Materials and methods
### Cloning and site-directed mutagenesis
Full-length human P-Rex1, Cdc42, and DH/PH-DEP1 expression constructs were described previously (**Cash et al., 2019**; **Cash et al., 2016**; **Ravala et al., 2020**). Mutations in DH/PH-DEP1 were created using QuikChange (QIAGEN) and confirmed by DNA sequencing. Mutations in the pCEFL-HA-HaloTag-P-Rex1 WT construct were created by QuikChange II site-directed mutagenesis (Agilent 200523). All constructs were confirmed by sequencing and expression was tested by immunoblot.

### Protein purification
Full-length P-Rex1 was transiently expressed in Freestyle 293-F cells and purified as discussed previously (**Cash et al., 2019**). Briefly, 48 hr after transfection, the cells were harvested and lysed with Cell Lytic M (Sigma). After ultracentrifugation to remove the insoluble fraction, the protein was purified using glutathione agarose resin (Gold Biotechnology Inc). The protein was subjected to TEV cleavage to remove the GST tag and then further purified using a Mono Q 5/50 GL anion exchange column (GE Healthcare Life Sciences). Finally, the protein was purified over an affinity column generated by conjugating human Rac1 to Affi-Gel 10 resin, although for Krios cryo-EM and HDX-MS experiments, this step was omitted.

P-Rex1 DH/PH-DEP1 proteins were expressed and purified as described previously (**Ravala et al., 2020**). Briefly, His-tagged protein was expressed in *Escherichia coli* BL21(DE3) cells, which were then lysed using an Avestin Emulsiflex-C3 high-pressure homogenizer. The cell lysate was clarified with high-speed centrifugation, the supernatant was collected, and protein was purified using Ni-NTA resin. The protein was subjected to TEV cleavage to remove the tag. The protein was then purified using a HiTrap SP sepharose column, concentrated, and subjected to size-exclusion chromatography on a Superdex S75 column (GE Healthcare) column. Cdc42 was produced in an unprenylated form in *E. coli* and purified as previously described (**Cash et al., 2016**).

### Hydrogen-deuterium exchange mass spectrometry
HDX-MS experiments were performed as previously described (**Cash et al., 2019**). Briefly, samples were mixed with $D_2O$ buffer to initiate the HDX reaction and, at various time points, the reaction was quenched with ice cold quench buffer and the samples immediately frozen on dry ice. Samples were thawed at 4°C and subjected to enzymatic digestion on an immobilized pepsin column followed by LC

separation and MS analysis. Data were analyzed using HDExaminer (Sierra Analytics, LLC, Modesto, CA). Each sample was analyzed twice by HDX-MS, and the data shown in the graphs of *Figure 1B and C* represent the average of these experiments. Each image in *Figure 4—source data 1* and *Figure 6—source data 1* shows one experiment (rainbow plots) or a difference analysis from those experiments (red to blue plots). Only one of the two sets of experiments performed for each condition (±liposomes or ± IP$_4$) are shown. For each peptide, the average of all five time points was calculated and used to plot the difference data onto the coordinates. Coordinates are colored using a range of –20% (darkest blue, protection) to 20% (darkest red, deprotection). P-Rex1 was used at a concentration of 1.7 mg/ml. IP$_4$ (Cayman Chemical) was added at a concentration of 100 μM. For experiments with liposomes, liposomes were added at a molar ratio of 1 P-Rex1 to 4000 total lipids. Liposomes were composed of 80:80:1 POPC:POPS:PIP$_3$ and prepared as previously described (*Cash et al., 2019*).

## Guanine-nucleotide exchange assays

Proteins were evaluated for their GEF activity using a fluorescence-based assay (*Cash et al., 2019*). Briefly, N-methyl-anthraniloyl-GDP (mant-GDP)-loaded soluble Cdc42 was used as a substrate GTPase (2 μM) in a buffer containing 20 mM HEPES pH 8, 100 mM NaCl, 0.5 mM MgCl$_2$, 100 μM GTP, and reactions were started by addition of P-Rex1 (100 nM). The loss of fluorescence was measured over time at 10 s intervals on a Flexstation 3 plate reader for 40 min. The data was fit to the one-phase exponential decay model in GraphPad Prism with the span (Y$_o$-plateau) shared among samples. For IP$_4$ competition curves, GEF assays were carried out in the presence of liposomes containing 2.5 μM PIP$_3$, as indicated, and 200 μM each of POPC and POPS, prepared as described previously (*Cash et al., 2019*).

## Cryo-EM grid preparation and data collection

For cryo-EM sample preparation, P-Rex1 was used at a final concentration of 3 μM and *n*-dodecyl-β-D-maltoside (DDM) was added to a final concentration of 0.08 mM. For samples with IP$_4$, a final concentration of 40 μM IP$_4$ was added. A sample of 4 μl was applied to a glow-discharged Quantifoil (1.2/1.3) 300-mesh grid, which was then blotted with filter paper and plunge-frozen into liquid ethane cooled with liquid nitrogen using a Vitrobot Mark IV (Thermo Fisher Scientific) set to 4°C, 100% humidity, 4 s blot, and a force of 10. Micrographs were collected either using Leginon (*Suloway et al., 2005*) on a Glacios transmission electron microscope (Thermo Fisher Scientific) operating at 200 keV and a K2 Summit direct electron detector (Gatan, Inc) in counting mode (0.98 Å/pixel) at a nominal magnification of ×45,000 or using EPU (Thermo Fisher Scientific) on a Titan Krios transmission electron microscope (Thermo Fisher Scientific) operating at 300 keV and a K3 direct electron detector (Gatan, Inc) in counting mode (1.054 Å/pixel) at a nominal magnification of ×81,000. On the Krios, datasets were collected on both untilted and 30° tilted grids (*Table 1*).

## Cryo-EM data processing

To overcome the severe preferred orientation problem of our sample on grids, we collected data on 0° and 30° tilted samples on a Krios electron microscope and processed these datasets separately up through 2D classification (*Table 1*). For each dataset, micrograph assessment, particle picking, and contrast transfer function estimation were performed using Warp (*Tegunov and Cramer, 2019*). Particle stacks were taken into CryoSPARC (*Punjani et al., 2020*; *Punjani et al., 2017*) and extensively cleaned using 2D classification. A final merged particle stack was used for ab initio reconstruction into one class followed by non-uniform refinement to obtain a map at an overall 4.1 Å resolution for the P-Rex1·IP$_4$ complex. For Glacios datasets, data were processed only through 2D classification.

## Model building and refinement

Initial model building relied on docking existing atomic models for the DH and PH domains of P-Rex1 (PDB entries 5FL1 and 5D3Y) and the Gβγ-binding scaffold (PDB entry 6PCV). 5D3Y was used for the PH domain because the maps were consistent with IP$_4$ bound to the PIP$_3$ site of the PH domain. The DEP1 domain was placed using a non-domain swapped atomic model derived from PDB entry 6VSK. The linker between the PH and DEP1 domains and the 4HB domain was built by hand. A Dali search (*Holm and Laakso, 2016*) using backbone helices of 4HB revealed its topology to be similar to FAT domains, which was then used to adjust the register of each of its four helices. When Alphafold2

(*Jumper et al., 2021*) became available, it was used to further adjust the modeling of the 4HB domain and associated structural elements in the IP4P domain. Finally, when the cryo-EM structure of P-Rex1 (PDB entry 7SYF) and the crystal structure of the DH/PH-DEP1 module (PDB entry 7*R*X9) became available, they were used to confirm less certain regions. Final rounds of real space refinement iterating with manual building were performed in Phenix (*Adams et al., 2019*). Final structure statistics are given in *Table 1*, and the structure and associated maps were deposited as PDB entry 8TUA and EMDB entry EMD-41621. Raw data were deposited as EMPIAR entry EMPIAR-11967.

## Structure visualization
UCSF ChimeraX (*Pettersen et al., 2021*) was used to make figures showing cryo-EM maps. PyMOL (The PyMOL Molecular Graphics System, version 2.5.5, Schrödinger, LLC) was used to create all other structure images.

## Size-exclusion chromatography coupled to small-angle X-ray scattering (SEC-SAXS)
For in-solution characterization of DH/PH-DEP1 and its variants, SEC-SAXS was conducted at the BioCAT beamline (Sector18) at the Advanced Photon Source, Argonne National Laboratory, using an AKTA Pure FPLC and a Pilatus3 X 1M detector. Purified proteins were injected onto a Superdex 200 Increase 10/300 column at a final concentration of 3–5 mg/ml in 20 mM HEPES, pH 7, 300 mM NaCl, 2% glycerol, and elution from this column flowed into SAXS flow cell for X-ray scattering. Data were collected every 1 s with 0.5 s exposure times at room temperature using 12 keV X-rays (1.033 Å wavelength) and a 3.67 m sample-to-detector distance. The achievable q range for this experimental setup was (0.0043–0.3546 Å).

## SAXS analysis
The scattering data were processed using BioXTAS RAW 1.6.3 software (*Hopkins et al., 2017*; *Figure 3—figure supplement 2A*) and used to determine the forward scattering I(0) and the radius of gyration, $R_g$ via Guinier analysis (*Figure 3—figure supplement 2B*). The Kratky plot showed proteins to be flexible (*Figure 3—figure supplement 2C*), leading to unsuccessful rigid modeling efforts. However, use of the EOM within ATSAS/3.0.5-2 aided in generating ensembles representing distinct conformational states of the DH/PH-DEP1 fragment and its variants A170K, L177E, and I409A in solution at equilibrium (*Ravala et al., 2020*). For EOM, the models were generated using crystallographic coordinates from their respective crystal structures: PDB 5FI1 and PDB 6VSK. EOM generated 50,000 possible profiles for the full pool using default settings and native-like structures. From these profiles, a sub-ensemble that matches the experimental scattering data is selected by a genetic algorithm run 10 times using default settings to verify the stability of the results (results from 1 run are shown in *Figure 3E and F*). The pair distance distribution function P(r), which provides maximum particle dimension of each protein, was calculated using GNOM (DI, 1992) from the ATSAS 2.8.4 package (*Franke et al., 2017*; *Figure 3—figure supplement 2D*).

For quantitative analyses of the flexibility of the selected ensembles, $R_{flex}$ and $R_\sigma$ metrics were derived from EOM data. WT DH/PH-DEP1 and A170K show $R_{flex}$ values smaller than those of the pools, which indicates that these proteins do not exhibit a fully flexible conformation. L177E and I409A variants exhibit $R_{flex}$ values close to those of the pool, suggestive of being highly flexible. Since Guinier analyses and the normalized residual fits of the proteins show that SAXS data quality is good (*Figure 3—figure supplement 2B*), the $R_\sigma$ value >1 is due to flexibility in the protein, consistent with the EOM analysis (*Figure 3E and F*, *Figure 3—figure supplement 2A*). Results of the SAXS analysis are presented in *Table 3* in accordance with the revised guidelines for publishing SAXS data (*Trewhella et al., 2017*). The SAXS data are deposited in SASBD (https://www.sasbdb.org/) with accession codes SASDUF2 for DH/PH-DEP1 WT, SASDUG2 for DH/PH-DEP1 A170K, SASDUH2 for DH/PH-DEP1 L177E, and SASDUJ2 for DH/PH-DEP1 I409A.

## Differential scanning fluorimetry
ThermoFluor experiments were performed on a QuantStudio 5 Real-Time PCR system in duplicate with n = 3 independent experiments. Purified DH/PH-DEP1 and its variants were incubated at 1 mg/ml in a buffer containing 20 mM HEPES pH 7.0, 300 mM NaCl, and 2 mM DTT with 2.5× Sypro Orange

dye in a final volume of 10 µl in a 384-well PCR plate. Fluorescence was monitored as a function of temperature, and $T_m$ was determined by fitting the fluorescence data to a sigmoidal curve and calculating the inflection point in GraphPad Prism.

### Luciferase-gene reporter assay

HEK293T cells seeded in 12-well plates coated with poly-D-lysine were transfected with 500 ng of empty vector pHTN HaloTag CMV-neo (Promega G7721) or pCEFL-HaloTag-P-Rex1 constructs and co-transfected with 500 ng of SRE-firefly luciferase and 50 ng of Renilla luciferase plasmids. Thirty-six hours after transfection, the cells were serum-starved overnight and then luminescence signal was measured using Dual-Glo assay system (Promega E2920) according to the manufacturer's instructions. Firefly-luminescence reads were normalized with Renilla-luminescence signal and adjusted to the negative control.

### Preparation of P-Rex1 KO in HeLa cells and cell migration assays

HeLa cells were lentiviral transduced with pLentiCRISPRv2 P-Rex1 guide RNA3 - AGGCATTCCTGC ATCGCATC (GenScript SC1678). Forty-eight hours after transduction, HeLa cells were selected with puromycin (3 µg/ml) for 7 days. P-Rex1 KO was confirmed by western blot. Chemotactic migration was measured by trans-well assays (Thermo Scientific 140629). Inserts of 24-well plates were pre-treated with fibronectin (50 µg/ml) for 3 hr at 37°C. Subsequently, $5 \times 10^4$ HeLa cells prepared in serum-free DMEM were plated on the inserts at the upper chamber. Human CXCL12 (50 ng/ml) and EGF (50 ng/ml) (Sigma-Aldrich SRP3276 and SRP6253) were prepared in serum-free DMEM and used as chemoattractant in the lower chamber. Serum-free DMEM was used as negative control. The plates with the inserts were incubated in a humidified atmosphere at 37°C and 5% $CO_2$ for 6 hr. After incubation, the cells at the upper surface of the membrane insert were carefully removed and the cells attached to the lower surface were gently washed with PBS and then fixed with 4% paraformaldehyde for 15 min. After fixation, the cells were gently washed with PBS followed by staining with 0.5% crystal violet for 20 min. Excess of crystal violet was removed by gentle PBS washes. Migrating cells were imaged using an inverted microscope. Quantification of particles corresponding to migrating cells was performed with FIJI software.

### Western blot

Protein samples prepared in Laemmli buffer were separated with SDS-PAGE using 4–12% gradient gels followed by transfer to PVDF membranes. The membranes were blocked using 5% non-fat milk in TBS-0.05% Tween20 (TBST) and incubated overnight at 4°C with primary antibodies against P-Rex1 and GAPDH (Cell Signaling Technology #13168 and #5174, respectively). The membranes were washed three times with TBST and then incubated with secondary antibodies in blocking solution for 2 hr at room temperature. After washing three times with TBST, the reactive bands were visualized using ECL detection reagents and CL-X-posure films.

### Statistical analysis

GEF assays described in this study were performed with n ≥ 3 replicates, with statistical significance determined using one-way ANOVA test with a post hoc Dunnett's test for multiple comparisons. Luciferase assays were conducted in triplicate, with 5 measurements for each variant, whereas chemotaxis experiments were from at least three independent measurements. In both cases, significance was determined using multiple comparison ANOVA followed by Šidák statistic test.

## Acknowledgements

We thank Dr. Srinivas Chakravarthy, Beamline Scientist, for performing SEC-SAXS at BioCAT, Chicago, IL, and Dr. Jesse B Hopkins, Deputy Director, BioCAT, for help with SAXS data analysis and interpretation. This research used resources of the Advanced Photon Source, a U.S. Department of Energy (DOE) Office of Science User Facility operated for the DOE Office of Science by Argonne National Laboratory under Contract No. DE-AC02-06CH11357. BioCAT was supported by grant P30 GM138395 from the National Institute of General Medical Sciences of the National Institutes of Health. We thank Dr. Thomas Klose in the Purdue Cryo-EM Facility for technical assistance in cryo-EM data collection. Research reported in this publication was supported by National Institutes of Health grants CA254402,

CA221289, HL071818, P30CA023168 (JJGT), the Walther Cancer Foundation (JJGT), and the National Institute of General Medical Sciences of the National Institutes of Health grant R35GM146664 (JNC).

## Additional information

### Funding

| Funder | Grant reference number | Author |
|---|---|---|
| National Institute of General Medical Sciences | R35GM146664 | Jennifer N Cash |
| Walther Cancer Foundation | Walther Professor of Cancer Structural Biology | John JG Tesmer |
| National Cancer Institute | CA254402 | Sandeep K Ravala<br>Sendi Rafael Adame-Garcia<br>Sheng Li<br>Chun-Liang Chen<br>J Silvio Gutkind<br>Jennifer Cash<br>John JG Tesmer |
| National Cancer Institute | CA221289 | Sandeep K Ravala<br>Sendi Rafael Adame-Garcia<br>Sheng Li<br>Chun-Liang Chen<br>J Silvio Gutkind<br>Jennifer N Cash<br>John JG Tesmer |
| National Heart, Lung, and Blood Institute | HL071818 | Sandeep K Ravala<br>Sendi Rafael Adame-Garcia<br>Sheng Li<br>Chun-Liang Chen<br>J Silvio Gutkind<br>Jennifer N Cash<br>John JG Tesmer |
| National Cancer Institute | P30CA023168 | Sandeep K Ravala<br>Chun-Liang Chen<br>Jennifer Cash<br>John JG Tesmer |

The funders had no role in study design, data collection and interpretation, or the decision to submit the work for publication.

### Author contributions

Sandeep K Ravala, Data curation, Formal analysis, Investigation, Writing – original draft, Writing – review and editing; Sendi Rafael Adame-Garcia, Formal analysis, Investigation, Writing – review and editing; Sheng Li, Formal analysis, Investigation; Chun-Liang Chen, Investigation; Michael A Cianfrocco, Resources, Data curation; J Silvio Gutkind, Funding acquisition, Project administration; Jennifer N Cash, Conceptualization, Data curation, Formal analysis, Funding acquisition, Investigation, Writing – original draft, Writing – review and editing; John JG Tesmer, Conceptualization, Formal analysis, Supervision, Funding acquisition, Writing – original draft, Project administration, Writing – review and editing

### Author ORCIDs

Chun-Liang Chen ⓘ https://orcid.org/0000-0003-4625-4340
Michael A Cianfrocco ⓘ https://orcid.org/0000-0002-2067-4999
Jennifer N Cash ⓘ https://orcid.org/0000-0002-0277-7652
John JG Tesmer ⓘ https://orcid.org/0000-0003-1125-3727

Reviewer #1 (Public Review): https://doi.org/10.7554/eLife.92822.4.sa1
Reviewer #2 (Public Review): https://doi.org/10.7554/eLife.92822.4.sa2
Reviewer #3 (Public Review): https://doi.org/10.7554/eLife.92822.4.sa3
Author response https://doi.org/10.7554/eLife.92822.4.sa4

## Additional files

### Supplementary files
• MDAR checklist

### Data availability

Data have been deposited to the PDB under accession code 8TUA, to the EMDB under accession code EMD-41621, and to EMPIAR under accession code EMPIAR-11967.

The following datasets were generated:

| Author(s) | Year | Dataset title | Dataset URL | Database and Identifier |
|---|---|---|---|---|
| Cash JN, Tesmer JJG | 2024 | Full-length P-Rex1 in complex with inositol 1,3,4,5-tetrakisphosphate (IP4) | https://www.rcsb.org/structure/8TUA | RCSB Protein Data Bank, 8TUA |
| Cash JN, Tesmer JJG | 2024 | Full-length P-Rex1 in complex with inositol 1,3,4,5-tetrakisphosphate (IP4) | https://www.ebi.ac.uk/emdb/EMD-41621 | EMDB, EMD-41621 |
| Cash JN, Tesmer JJG | 2024 | Cryo-EM structure of P-Rex1-IP4 | https://www.ebi.ac.uk/empiar/EMPIAR-11967/ | EMPIAR, EMPIAR-11967 |

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
