## [Editor Report · eLife assessment]

This **important** study contributes insights into the regulatory mechanisms of a protein governing cell migration at the membrane. The integration of approaches revealing protein structure and dynamics provides **convincing** data for a model of regulation and suggests a new allosteric role for a solubilized phospholipid headgroup. The work will be interesting to researchers focusing on signaling mechanisms, cell motility, and cancer metathesis.

---

## [Referee Report · Reviewer #1 (Public Review)]

Summary:

The authors perform a multidisciplinary approach to describe the conformational plasticity of P-Rex1 in various states (autoinhibited, IP4 bound and PIP3 bound). Hydrogen-deuterium exchange (HDX) is used to reveal how IP4 and PIP3 binding affect intramolecular interactions. While IP4 is found to stabilize autoinhibitory interactions, PIP3 does the opposite, leading to deprotection of autoinhibitory sites. Cryo-EM of IP4 bound P-Rex1 reveals a structure in the autoinhibited conformation, very similar to the unliganded structure reported previously (Chang et al. 2022). Mutations at observed autoinhibitory interfaces result in a more open structure (as shown by SAXS), reduced thermal stability and increased GEF activity in biochemical and cellular assays. Together their work portrays a dynamic enzyme that undergoes long-range conformational changes upon activation on PIP3 membranes. The results are technically sound and the conclusions are justified. The main drawback is the limited novelty due to the recently published structure of unliganded P-Rex1, which is virtually identical to the IP4 bound structure presented here. Novel aspects suggest a regulatory role for IP4, but the exact significance and mechanism of this regulation has not been explored.

Strengths:

The authors use a multitude of techniques to describe the dynamic nature and conformational changes of P-Rex1 upon binding to IP4 and PIP3 membranes. The different approaches together fit well with the overall conclusion that IP4 binding negatively regulates P-Rex1, while binding to PIP3 membranes leads to conformational opening and catalytic activation. The experiments are performed very thoroughly and are technically sound. The results are clear and support the conclusions.

Weaknesses:

(1) The novelty of the study is compromised due to the recently published structure of unliganded P-Rex1 (Chang et al. 2022). The unliganded and IP4 bound structure of P-Rex1 appear virtually identical, however, no clear comparison is presented in the manuscript. In the same paper a very similar model of P-Rex1 activation upon binding to PIP3 membranes and Gbeta-gamma is presented.

(2) The authors demonstrate that IP4 binding to P-Rex1 results in catalytic inhibition and increased protection of autoinhibitory interfaces, as judged by HDX. The relevance of this in a cellular setting is not clear and is not experimentally demonstrated. Further, mechanistically, it is not clear whether the biochemical inhibition by IP4 of PIP3 activated P-Rex1 is due to competition of IP4 with activating PIP3 binding to the PH domain of P-Rex1, or due to stabilizing the autoinhibited conformation, or both.

---

## [Referee Report · Reviewer #2 (Public Review)]

Summary:

In this new paper, the authors used biochemical, structural, and biophysical methods to elucidate the mechanisms by which IP4, the PIP3 headgroup, can induce an autoinhibit form of P-Rex1 and propose a model of how PIP3 can trigger long-range conformational changes of P-Rex1 to relieve this autoinhibition. The main findings of this study are that a new P-Rex1 autoinhibition is driven by an IP4-induced binding of the PH domain to the DH domain active site and that this autoinhibit form stabilized by two key interactions between DEP1 and DH and between PH and IP4P 4-helix bundle (4HB) subdomain. Moreover, they found that the binding of phospholipid PIP3 to the PH domain can disrupt these interactions to relieve P-Rex1 autoinhibition.

Strengths:

The study provides good evidence that binding of IP4 to the P-Rex1 PH domain can make the two long-range interactions between the catalytic DH domain and the first DEP domain, and between the PH domain and the C-terminal IP4P 4HB subdomain that generate a novel P-Rex1 autoinhibition mechanism. This valuable finding adds an extra layer of P-Rex1 regulation (perhaps in the cytoplasm) to the synergistic activation by phospholipid PIP3 and the heterotrimeric Gβγ subunits at the plasma membrane. Overall, this manuscript's goal sounds interesting, the experimental data were carried out carefully and reliably.

Weakness:

The set of experiments with the disulfide bond S235C/M244C caused a bit of confusion for interpretation, it should be moved into the supplement, and the text and Figure 4 were altered accordingly.

---

## [Referee Report · Reviewer #3 (Public Review)]

Summary:

In this report, Ravala et al demonstrate that IP4, the soluble head-group of phosphatiylinositol 3,4,5 - trisphosphate (PIP3), is an inhibitor of pREX-1, a guanine nucleotide exchange factor (GEF) for Rac1 and related small G proteins that regulate cell cell migration. This finding is perhaps unexpected since pREX-1 activity is PIP3-dependent. By way of Cryo-EM (revealing the structure of the p-REX-1/IP4 complex at 4.2Å resolution), hydrogen-deuterium mass spectrometry and small angle X-ray scattering, they deduce a mechanism for IP4 activation, and conduct mutagenic and cell-based signaling assays that support it. The major finding is that IP4 stabilizes two interdomain interfaces that block access of the DH domain, which conveys GEF activity towards small G protein substrates. One of these is the interface between the PH domain that binds to IP4 and a 4-helix bundle extension of the IP4 Phosphatase domain and the DEP1 domain. The two interfaces are connected by a long helix that extends from PH to DEP1. Although the structure of fully activated pREX-1 has not been determined, the authors propose a "jackknife" mechanism, similar to that described earlier by Chang et al (2022) (referenced in the author's manuscript) in which binding of IP3 relieves a kink in a helix that links the PH/DH modules and allows the DH-PH-DEP triad to assume an extended conformation in which the DH domain is accessible. While the structure of the activated pREX-1 has not been determined, cysteine mutagenesis that enforces the proposed kink is consistent with this hypothesis. SAXS and HDX-MS experiments suggest that IP4 acts by stiffening the inhibitory interfaces, rather than by reorganizing them. Indeed, the cryo-EM structure of ligand-free pREX-1 shows that interdomain contacts are largely retained in the absence of IP4.

Strengths:

The manuscript thus describes a novel regulatory role for IP4 and is thus of considerable significance to our understanding of regulatory mechanisms that control cell migration, particularly in immune cell populations. Specifically, they show how the inositol polyphosphate IP4 controls the activity of pREX-1, a guanine nucleotide exchange factor that controls the activity of small G proteins Rac and CDC42. In their clearly-written discussion, the authors explain how PIP3, the cell membrane and the Gbeta-gamma subunits of heterotrimeric membranes together localize pREX-1 at the membrane and induce activation. The quality of experimental data is high and both in vitro and cell-based assays of site-directed mutants designed to test the author's hypotheses are confirmatory. The results strongly support the conclusions. The combination of cryo-EM data, that describe the static (if heterogeneous) structures with experiments (small angle x-ray scattering and hydrogen-deuterium exchange-mass spectrometry) that report on dynamics are well employed by the authors

Manuscript revision:

The reviewers noted a number of weaknesses, including error analysis of the HDX data, interpretation of the mutagenesis data, the small fraction of the total number of particles used to generate the EM reconstruction, the novelty of the findings in light of the previous report by Cheng et al, 2022, various details regarding presentation of structural results and questions regarding the interpretation of the inhibition data (Figure 1D). The authors have responded adequately to these critiques. It appears that pREX-1 is a highly dynamic molecule, and considerable heterogeneity among particles might be expected.

While, indeed, the conformation of pREX presented in this report is not novel, the finding that this inactive conformational state is stabilized by IP4 is significant and important. The evidence for this is both structural and biochemical, as indicated by micromolar competition of IP4 with PI3-enriched vesicles resulting in the inhibition of pREX-1 GEF activity.

---

## [Author Response]

The following is the authors’ response to the previous reviews.

We thank the reviewers for their thorough review of and overall positive comments on our manuscript. We have revised the manuscript to address the one remaining concern raised by one of the reviewers. This is described below.

Fig.1B-C: To give a standard deviation from 2 data points has no statistical significance. In this case it would be better to define as range/difference of the 2 data points.

We have modified the legend for Figure 1 to now read, “The average of two experiments is plotted with the bars representing the range of each time point.”